# Pharmacological markers of HIV prevention for oral pre-exposure prophylaxis in men who have sex with men

Sara Iannuzzi[1,2], Malin Müller [1], Yifan Yu[3], Lanxin Zhang[1,2], Craig W. Hendrix [4], Robert R. Bies[3] & Max von Kleist [1,2] ✉

The human immunodeficiency virus (HIV) infected approximately 1.1 million individuals in 2024. There is no effective vaccine or cure, and funding cuts in resource-limited settings threaten treatment access. Cost-effective and widely available prevention strategies, such as oral emtricitabine/tenofovir disoproxil fumarate pre-exposure prophylaxis (FTC/TDF-PrEP), are therefore essential. Current PrEP guidelines differ between cisgender women and men who have sex with men (MSM), based on mechanistic differences in tissue-level pharmacokinetics (PK) at vaginal vs. colorectal exposure sites. To test these mechanistic hypothesis, we use data from major FTC/TDF-PrEP trials to establish PrEP efficacy when used in MSM. We independently predict efficacy utilizing different PK-matrices in a mechanistic model, simulate each clinical trial informed by adherence data and compare the predictions with clinical efficacy estimates. With this combined approach, two of the five trials (HPTN 083, DISCOVER) yield sufficient statistical power to conclude that rectal tissue pharmacokinetics do not predict PrEP efficacy in MSM. In contrast, PBMC-based predictions agree with clinical PrEP efficacy and support the suitability of on-demand use of oral PrEP in MSM. When combining our findings with recent results on suitable pharmacokinetic markers in women, our work suggests that adherence requirements for cisgender women and MSM may not differ.

Approximately 2-3 individuals are infected with the human immunodeficiency virus (HIV-1) every minute.[1] Virus suppression with combinations of antiretroviral drugs, known as highly active antiretroviral treatment (HAART), prevents progression to acquired immunodeficiency syndrome (AIDS) and death[2], as well as preventing HIV transmission[3].

Unfortunately, numerous vaccine trials have failed to demonstrate highly effective HIV prevention[4,5]. In the absence of a vaccine option, several antivirals in a variety of dosage formulations are highly efficacious as pre-exposure prophylaxis (PrEP) to prevent HIV infection.

Most recently, twice-yearly injections with long-acting lenacapavir demonstrated greater than 90% PrEP efficacy[6,7]. However, long-acting agents remain unavailable or cost-prohibitive across much of the globe[8,9], particularly as major funding programs are being stopped. In contrast, daily PrEP with tenofovir disoproxil fumarate + emtricitabine (TDF/FTC) is widely available and used around the globe[10] and is available in several generic formulations providing a highly effective

[1]Projektgruppen, Robert-Koch Institute Berlin, Berlin, Germany. [2]Department of Mathematics and Computer Science, Freie Universität Berlin, Berlin, Germany. [3]Division of Pharmacokinetics, Pharmacodynamics and Systems Pharmacology, School of Pharmacy and Pharmaceutical Sciences, University at Buffalo, Buffalo, NY, USA. [4]Department of Medicine (Clinical Pharmacology) and Pharmacology and Molecular Sciences, Johns Hopkins University School of Medicine, Baltimore, MD, USA. ✉e-mail: max.kleist@fu-berlin.de

and cost-efficient option to prevent HIV transmission[11], when taken as prescribed.

The WHO and US PrEP guidelines recommend different oral PrEP regimens for cisgender women and men who have sex with men (MSM)[12,13], whereas the 2025 UK guidelines do not make a distinction[14]. According to the WHO and US guidelines MSM can take PrEP on demand shortly before sexual activity. These recommendations are supported by clinical studies that found no statistical difference between daily- and on demand regimens in MSM[15,16]. Notably, no such study has been completed in women. A recent systematic comparison of PrEP efficacy and adherence benchmarks from several large clinical studies indicated that ≥4 doses per week provide protection in cisgender women, which is similar to MSM.[17] In contrast, a pooled analysis of three major clinical studies postulated that women at high risk of acquiring HIV required higher protective TFV plasma concentrations than men at high risk, albeit with no notable gender-specific differentiation when the risk of acquiring HIV was low.[18] Historically, two simultaneous trends led some to postulate that women required higher levels of adherence than men: Based upon data from PrEP efficacy trials, PrEP treatment among women seemed to underperform in some trials[19,20], compared to men.[21] However, this trend was not consistent across all trials.[22] In parallel, accumulating clinical studies with tissue pharmacokinetic data showed differences in active intracellular TFV diphosphate (TFV-DP) and FTC triphosphate (FTC-TP) concentrations between rectal and cervico-vaginal tissues. These, in turn, motivated modeling to estimate TDF/FTC PrEP efficacy from TFV-DP and FTC-TP tissue concentrations. Using mixed clinical sample and in vitro experiments, Cottrell et al.[23] proposed that, because the concentration of active drugs are lower in vaginal tissue compared to rectal tissue, women require more frequent dosing (higher adherence demands) to achieve sufficient drug concentrations to protect from HIV. By combining an analysis of all major TDF/FTC-based oral PrEP studies in women with mechanistic systems pharmacological modeling, we showed recently[24] that drug concentrations in vaginal tissue cannot predict prophylactic efficacy in women. Moreover, we showed that drug concentrations in systemic peripheral blood mononuclear cells (PBMC), which mainly consist of HIV target (CD4+) cells, are highly predictive of clinical PrEP efficacy. Based thereon, we independently estimated that ≥4 doses per week are highly ( > 90%) efficient against receptive vaginal HIV transmission, in agreement with several recent analyses.[25,26] However, our previous analysis was not concluded by a corresponding analysis of pharmacological markers for MSM.

In this work, we investigate for MSM whether PBMC, which disregard any colorectal-vaginal differences, predict oral prophylactic efficacy. We utilize two independent approaches: (i) We establish ranges of PrEP efficacy when used for TDF/FTC-based oral PrEP from all major clinical trials in MSM, using Bayesian inference. (ii) We then use mechanistic modeling to test hypotheses regarding pharmacological markers of PrEP efficacy, by combining viral and immune dynamics with population pharmacokinetics to simulate the clinical trials in MSM. This way, clinical PrEP-efficacy estimates (from approach i) can be used to test hypotheses regarding pharmacological efficacy markers, utilized in the mechanistic modeling approach (approach ii). Finally, we use the non-rejected models to simulate and test adherence-efficacy relationships for oral PrEP and to assess how quickly protection is achieved in MSM and how long it lasts after stopping TDF/FTC-based oral PrEP.

## Results

### Clinical PrEP efficacy ranges in MSM with evidence of drug intake

Clinical trials with oral PrEP may entail individuals in the intervention arm who did not take the prescribed drugs at all, or for a period of time. In order to allow for a comparison between trials, which may

substantially differ with regard to the fraction of non-adherers, any observation time without detectable drug, as well as infections without detectable drug, was cleared from the intervention arm. This augmented data set was used to estimate average PrEP efficacy when used (subsequently termed PrEP efficacy), i.e., excluding individuals with no PK marker evidence of drug. Other than excluding individuals with no PK marker evidence of drug, average PrEP efficacy is calculated akin to a clinical trial, by estimating the reduction in incidence (which averages over trial-specific adherence and individual pharmacokinetics). By implementing population-pharmacokinetic models, we have previously shown (Supplementary Fig. S1 in ref. 24) that individuals with undetectable plasma tenofovir (TFV) levels took PrEP less than once per week (if at all), serving as a marker of recent adherence for earlier studies (IPERGAY, iPrEx, HPTN 083). To the contrary, more recent trials (PURPOSE 2, DISCOVER) established TFV-DP in red blood cells (RBCs) as a marker of adherence. Based on available markers (overview in Supplementary Table 3), we were able to dichotomize the observation time, as well as the number of infections in the oral FTC/TDF PrEP intervention arm into PrEP-taking (with some unknown adherence) vs. PrEP non-taking. Based on available adherence markers (plasma TFV and TFV-DP in dried blood spots (DBS) and PBMC, plasma FTC and FTC-TP in PBMC), 14%, 49%, 21%, 14% and 4% of participant samples in IPERGAY, iPrEX, PURPOSE 2, HPTN 083 and DISCOVER respectively indicated that PrEP was not recently taken (undetectable plasma TFV), or taken no more than once a week (based on DBS TFV-DP levels). The total number of infected individuals in the respective intervention arms was also dichotomized by the above indicated adherence markers. Both infections observed in the once-daily oral TDF/FTC PrEP intervention arm of IPERGAY had undetectable plasma TFV.[15] In iPrEx, 31 out of 34 observed infections were associated with undetectable plasma TFV.[21] The remaining 3 individuals had detectable levels. Similarly, in HPTN 083, 33 out of 39 infected individuals lacked evidence of recent drug intake based on both plasma TFV and TFV-DP in RBCs, while the remaining 6 individuals had evidence of some drug intake.[27] Of these, 4 individuals had TFV-DP RBC levels above the lower limit of quantification but below 350 fmol/punch, one had levels between ≥700 and < 1200 fmol/punch and one had levels ≥1200 fmol/punch (Supplementary Table 2). In PURPOSE 2, 6 out of 8 infected individuals, for whom samples were available, exhibited undetectable drug levels[6], the remaining 2 had TFV-DP levels in DBS corresponding to < 2 doses per week. In DISCOVER, 10 out of 11 infected individuals had TFV-DP levels in DBS corresponding to < 2 doses per week, while one individuals had levels corresponding to ≥ 4 doses per week.[28] A comparison of these findings is illustrated in Fig. 1a–e. To verify the appropriateness of this dichotomization, we compared the HIV incidences (number of infections/observation time) in the respective placebo arms (IPERGAY, iPrEX) with the incidences computed from the sub-group of the intervention arm, where plasma TFV was undetectable. The computed incidences generally agreed in magnitude (mean incidences fall within the 95% CI of the placebo group), however they also displayed large uncertainty, owed to smaller numbers of infections and shorter total observation time compared to the placebo arms (Fig. 1f). We could conclude from this data exploration, that it may be safe to assume 0% PrEP efficacy in individuals with undetectable TFV on the one hand, and that we have to consider uncertainty in infection incidence on the other hand. Consequently, when simulating the clinical trials, we took both the uncertainty in infection incidence, as well as intrinsic randomness due to small number of infection events into account, see Methods section.

Finally, we used an uninformative prior of PrEP efficacy, simulated the drug-detected intervention arm and computed a posterior efficacy by utilizing the distribution of data-derived infection numbers, see Methods section for details. This procedure allowed us to estimate average PrEP efficacy in individuals with evidence of drug intake without having to make assumptions about their actual

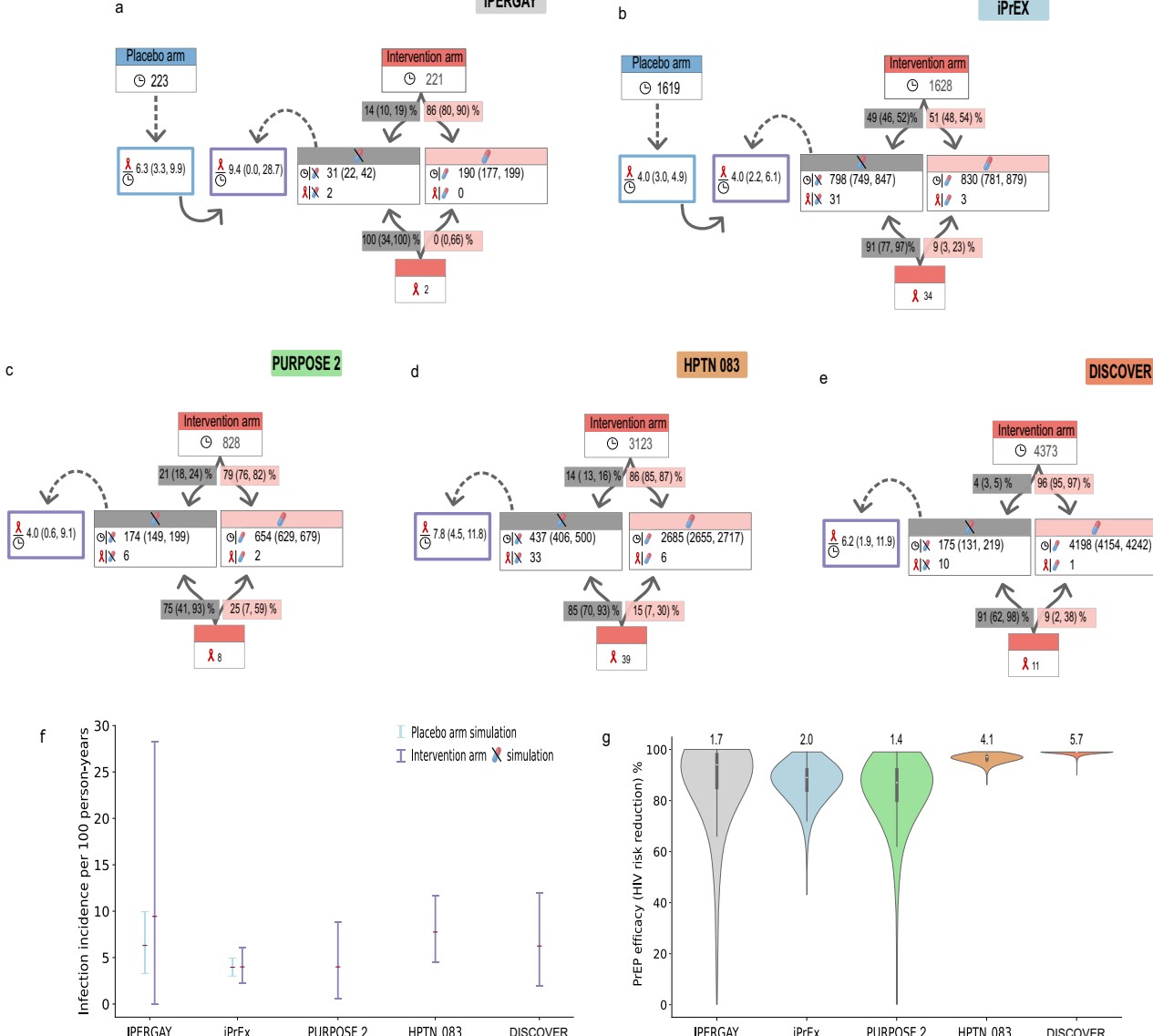

**Fig. 1 | Summary of clinical trials evaluating FTC/TDF-based PrEP and their predicted PrEP efficacy (when taken). a–e** The total observation time (in person-years, indicated by the clock) in the intervention arms was dichotomized based on the percentage of available samples where PrEP drug was detected (pill symbol, red shading) vs. not (crossed pill, grey shading). Percentages denote mean values with 95% Wilson CI in brackets.The total number of infections (red ribbon) was similarly dichotomized depending on whether plasma TFV and/or TFV-DP in DBS was detectable in infected individuals at their first HIV positive test. The dashed arrow indicates simulations performed using either the placebo data (blue) or, when placebo data were unavailable, the dichotomized drug-undetected arm (purple outline).**f** Mean (and 95% confidence ranges) of infection incidences ($r_{Inf}$) for placebo (where available, in blue) or sub-groups in the intervention arm where TFV was undetectable (in purple), with sample sizes indicated in **a–e**. The error bars in the placebo groups were estimated using Wilson's method. The purple bars are

based on both uncertainty in the incidence rate, as well as intrinsic randomness in trial outcomes due to rare infection events and are computed from 10,000 stochastic simulations. **g** The violin plots visualize uncertainty in average PrEP efficacy estimates for each study (boxplot: dots = median, box = IQR; whiskers = 95% CI), obtained by Bayesian inversion (see Methods), solely based on data from the studies (**a–e**). Horizontally stretched violin plots indicate greater certainty in average efficacy estimates, while vertically stretched violin plots indicate more uncertainty. We expressed this information gain from analyzing the respective clinical trial data as deviation from maximum Shannon entropy next to each violin (the higher the number, the more informative; see Methods section). Source data are provided as a Source Data file. FTC emtricitabine, TDF tenofovir disoproxil fumarate, PrEP pre-exposure prophylaxis, CI confidence interval, DBS dried blood spot, TFV tenofovir, TFV-DP tenofovir diphosphate.

adherence. Moreover, we could quantify the (un-) certainty associated with average PrEP efficacy for each clinical trial, which is depicted in Fig. 1g: The violin plots therein depict the uncertainty in estimating an average PrEP efficacy from the respective clinical data. A sharply concentrated distribution highlights certainty, whereas a wider distribution reflects uncertainty associated with an estimate of average PrEP efficacy, and a uniform distribution would indicate that the clinical trial does not entail any information regarding PrEP efficacy.

Notably, the dichotomization procedure enabled a cross-trial comparison of PrEP efficacy when used. The most likely average PrEP efficacy strata was (90–100%) in individuals with evidence of PrEP intake, across all studies (Fig. 1g). However, there are large differences regarding the information content provided by the distinct studies. HPTN 083 and DISCOVER enabled a more precise estimate of the average PrEP efficacy, allowing to state that average PrEP efficacy is equal to 90% or higher. In contrast, only weaker statements could be made from the iPrEx, IPERGAY and PURPOSE 2 clinical data, with a 37%,

63% and 35% probability that efficacy exceeded 90%. The data-derived uncertainty can be clearly attributed to less observation time in the sub-group with detectable TFV in iPrEx, IPERGAY and PURPOSE 2, possibly making these studies underpowered for elucidating pharmacological markers of oral PrEP efficacy (compare Fig. 1a–e). The derived information gain from analysis of the clinical data was quantified in terms of the deviation from a uninformative prior (numbers stated in Fig. 1g; details in Methods section), clearly indicating that HPTN 083 and DISCOVER allowed the most pronounced update of our non-informative prior towards a posterior estimate of average PrEP efficacy.

## Mechanistic modeling of PrEP efficacy against receptive anal intercourse

The clinical data alone does not enable us to test hypotheses regarding pharmacological markers associated with PrEP efficacy. Henceforth, we utilized a validated computational model of viral dynamics[29–31], which allowed us to integrate mechanism of action (MOA) models of direct drug effect for FTC-triphosphate (FTC-TP) and tenofovir-diphosphate (TFV-DP)[32] with population pharmacokinetic models[33,34] that relate arbitrary adherence patterns to target site concentrations.[24] Using the methods developed in ref. 35, we can use these integrated models to estimate PrEP efficacy for any virus exposure time and adherence pattern. Since precise adherence patterns are unknown to us, we used randomized adherence patterns for any tested average adherence strata: For example, an average of 3 doses per week could also imply an adherence pattern where 6 doses were taken in the first week and none in the second week. This may, in addition to pharmacokinetic variability, increase variability in estimated PrEP efficacy, because virus exposure may occur either at a time when many doses were taken (high protection), or when there was a gap in PrEP intake. Thus derived PrEP efficacies can then be used to simulate the clinical trials and evaluate the mechanistic predictions with regards to efficacy estimates derived from the clinical data (Fig. 1g). Overall, this approach allows us to rigorously test hypothesized pharmacological markers of PrEP efficacy.

We predicted PrEP efficacy based on the hypothesis that TFV-DP and FTC-TP concentrations in rectal tissue represent the effect compartment and compared these predictions to those derived using peripheral blood mononuclear cells (PBMC) as the effect compartment. In addition, we tested scenarios where the concentrations in the effect compartment lie in between tissue and PBMC concentrations (Supplementary Text, Supplementary Table S4).

To annotate the hypothesis testing scheme, we introduce the light switch shown in Fig. 2a.

Initially, to assess parameter sensitivity, we tested hypothesis in isolation. For 100% adherence to oral TDF/FTC PrEP, we predicted a mean efficacy of 98.4% (median: 99.0%), if drug concentrations in PBMC were used to estimate PrEP efficacy. When we used TFV-DP and FTC-TP concentrations in rectal tissue, we predicted a markedly reduced PrEP efficacy (mean 81.9%, median: 87.9%), which also displayed a wider range, which was associated with pharmacokinetic variability in this modeling approach. If we considered drug concentrations in PBMC, but computed drug potency ($IC_{50}$) using rectal dNTP concentrations, efficacy remained high at a mean of 99.5% (median: 100.0%) for 100% adherent individuals. This analysis was performed because TFV-DP and FTC-TP are competitive inhibitors whose molecular potency depends on dNTP concentrations[36] and it is unclear whether dNTP concentrations in tissue resident HIV target cells may be distinct from systemic cells (e.g. PBMC). Our analysis revealed that the choice of effect-site compartment (rectal vs. PBMC) denotes the most critical parameter, while there is little sensitivity with regards to dNTP levels. Henceforth, we utilized the dNTP concentrations closest to the drug concentration marker (PBMC & CD4+; rectal & rectal).

Next, we simulated the adherence-efficacy relationships for receptive anal intercourse (RAI) for the mechanistic hypothesis where drug concentrations in rectal tissue were used to predict efficacy (Fig. 2c, dark blue boxplots) versus a hypothesis where drug concentrations in systemic PBMC were used to predict PrEP efficacy (Fig. 2d, light blue). For the local tissue hypothesis, we observed that there is a gradual increase of PrEP efficacy with adherence and that protection would be incomplete even in individuals with full adherence. In contrast, when predicting PrEP efficacy against RAI based on TFV-DP & FTC-TP concentrations in PBMC, greater than 90% efficacy is already reached after 3–4 doses per week with only marginal increase for higher adherence. In the Supplementary Text and Supplementary Figs. 4, 5, we also modeled scenarios where 60% of exposures occurred via insertive anal intercourse (IAI), concluding that prophylactic efficacy against RAI by-and-large predicts prophylactic efficacy for MSM with both routes of exposure.

Next, we wanted to investigate how the choice of pharmacological marker (local tissue vs. PBMC) may relate to gender-specific differences in adherence-efficacy relationships. We therefore additionally simulated vaginal virus exposures using the mathematical model and assumed that either drug concentrations in local tissue (Fig. 2c, pink boxplots) vs. PBMC (Fig. 2d, light pink boxplots) predicted PrEP efficacy. We observed that if drug concentrations in local tissue predicted PrEP efficacy, there would be distinct adherence-efficacy relationships depending on whether an individual is virally exposed via the rectal, or vaginal route and that PrEP would generally be more effective against RAI. When predicting PrEP efficacy based on TFV-DP & FTC-TP concentrations in PBMC, we obtain indistinguishable adherence-efficacy profiles for RAI vs. RVI, albeit assuming a higher average inoculum size for rectal exposures in our simulations.

## Suitability of pharmacological matrices for predicting clinical PrEP efficacy

We wanted to test whether drug concentrations in the colorectal tissues or in the PBMC, generated using mechanistic modeling and simulation, predicted clinical PrEP efficacy estimates. For the respective trials, we sampled individual dosing patterns according to reported adherence data from the original studies (HPTN 083, PURPOSE 2, DISCOVER) or assumed adherence profiles based on plasma TFV detectability (IPERGAY, iPrEX), (Supplementary Fig. 1), and used respective pharmacokinetic matrices (colorectal tissue vs. PBMC) to estimate HIV risk reduction (PrEP efficacy). The thus derived simulated average PrEP efficacy was ≥90% for HPTN 083, IPERGAY, iPrEX, DISCOVER and PURPOSE 2, whenever PBMC was utilized to predict efficacy (Fig. 3, center row), whereas it was 73-82% whenever concentrations in colorectal tissues were used along with local dNTP levels (Fig. 3, bottom row).

We then stochastically simulated each clinical trial (utilizing its specific incidence, follow-up and adherence distribution) for the different hypotheses regarding drug concentrations matrices (colorectal vs. PBMC) and drug potency (dNTP levels). From these simulations, we could generate a probability distribution regarding possible trial outcomes in terms of PrEP efficacy (relative incidence reduction) and the number of infected individuals (see Methods section). We compared the obtained distributions with the corresponding distributions obtained directly from the dichotomized intervention arm (i.e. with detectable drug) of the clinical studies (Fig. 3). This allowed us to rigorously test whether the utilization of a specific drug concentration matrix in the mechanistic modeling may be inappropriate to predict clinical efficacy. For IPERGAY, iPrEX and PURPOSE 2 uncertainties in the clinical trial simulation outcome were too large (see also Fig. 1g) to distinguish mechanistic PrEP efficacy predictions utilizing colorectal tissue vs. PBMC drug concentrations. These studies are underpowered for our purposes, because of too little observation time in the sub-group of individuals with detectable drug (190, 830, 654 person-years

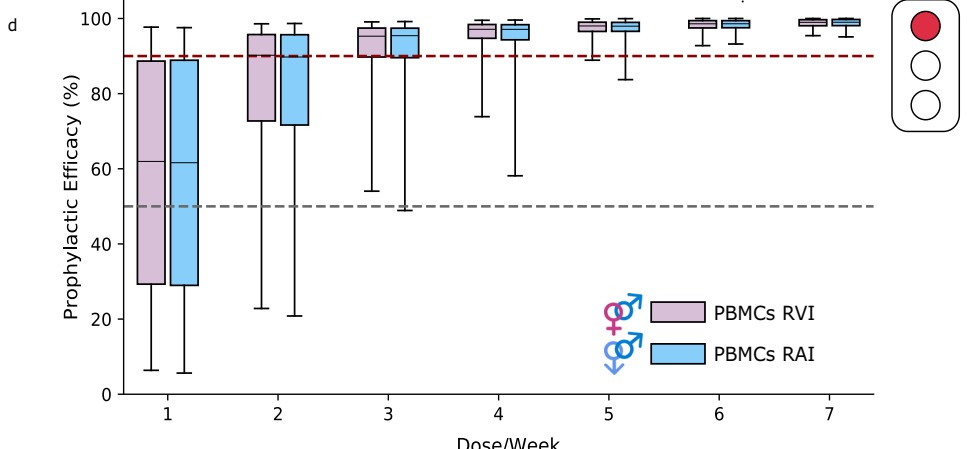

**Fig. 2 | Simulated PrEP efficacy and adherence-efficacy profiles using mechanistic modeling. a** Hypothesis testing scheme. Each color represents a hypothesis simulated in the mechanistic modeling framework. **b** Predicted PrEP efficacy against receptive anal intercourse considering hypotheses tested in isolation, under 100% dosing adherence. **c–d** Predicted PrEP adherence-efficacy relationship for vaginal (pink shade) vs. anal receptive intercourse (blue shade), when local (vaginal vs. colorectal) drug concentrations and dNTP levels were used to predict PrEP efficacy (**c**), vs. when drug concentrations and dNTP levels in PBMC were considered for predicting PrEP efficacy after anal- (light blue shade) vs. vaginal receptive virus exposure during sex (light pink shade, **d**). Central lines in the box-plots represent the median PrEP efficacy, boxes depict the inter-quartile ranges, and whiskers encompass the 95% percentile ranges when considering inter-individual pharmacokinetic variability in 1000 virtual individuals, based on 144,000 time points. Source data are provided as a Source Data file. PrEP: pre-exposure prophylaxis, PBMC peripheral blood mononuclear cells, dNTP deoxynucleoside triphosphate.

respectively) and henceforth, these clinical trial simulations could not be used to statistically rule out hypotheses at the $\alpha = 0.05$ level (columns: IPERGAY, iPrEX and PURPOSE 2 in Fig. 3). For HPTN 083 and DISCOVER, which encompasses a much larger observation period in

individuals with at least one FTC/TDF dose per week (2685 and 4198 person-years respectively) we could clearly rule out $P < 0.01$ that colorectal drug concentrations can be used to predict PrEP efficacy, (columns: HPTN 083 and DISCOVER in Fig. 3).

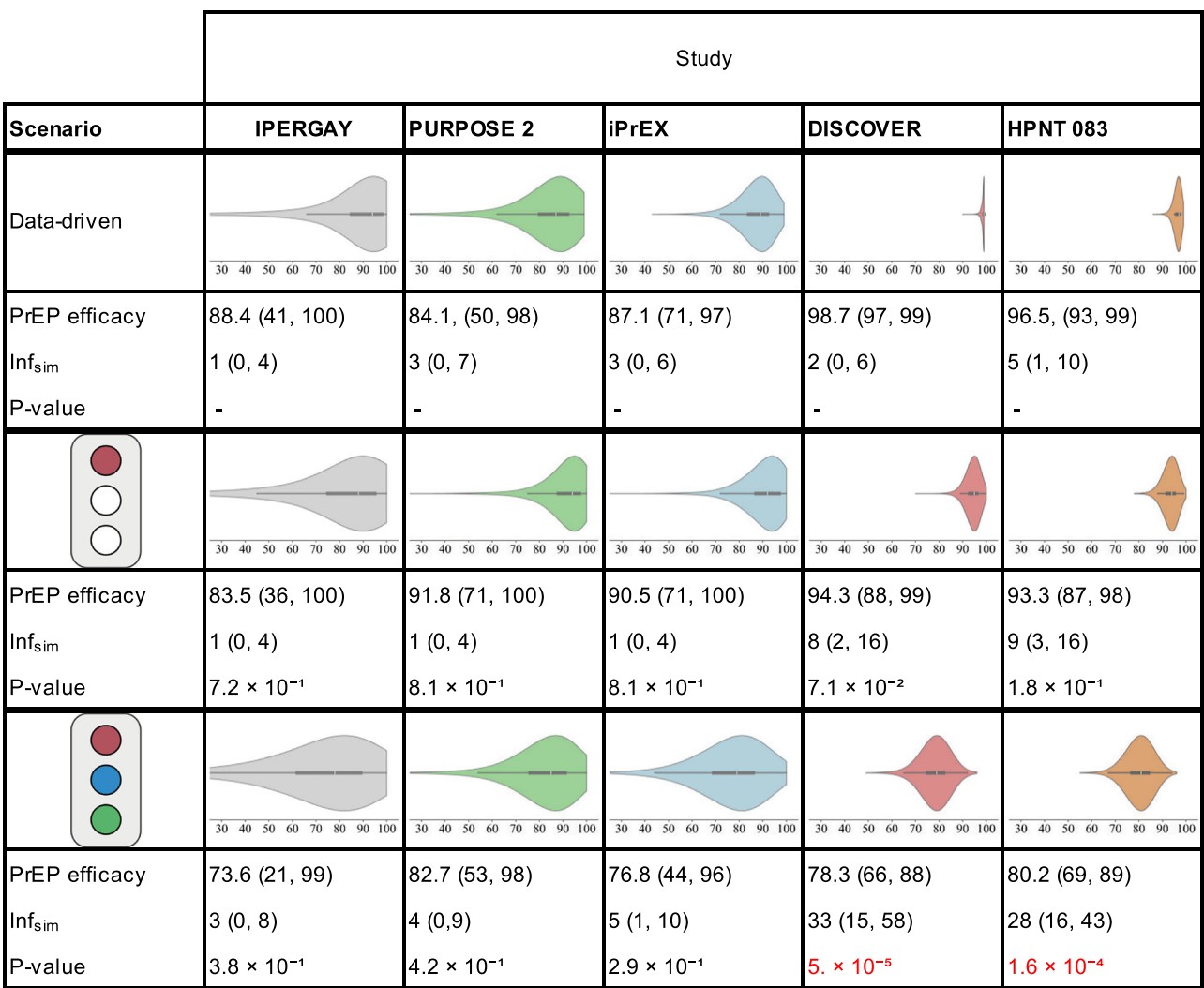

| Scenario | Study | | | | |
|---|---|---|---|---|---|
|  | **IPERGAY** | **PURPOSE 2** | **iPrEX** | **DISCOVER** | **HPNT 083** |
| Data-driven | (violin plot) | (violin plot) | (violin plot) | (violin plot) | (violin plot) |
| PrEP efficacy | 88.4 (41, 100) | 84.1, (50, 98) | 87.1 (71, 97) | 98.7 (97, 99) | 96.5, (93, 99) |
| Inf$_{sim}$ | 1 (0, 4) | 3 (0, 7) | 3 (0, 6) | 2 (0, 6) | 5 (1, 10) |
| P-value | - | - | - | - | - |
| (traffic light: red top, white) | (violin plot) | (violin plot) | (violin plot) | (violin plot) | (violin plot) |
| PrEP efficacy | 83.5 (36, 100) | 91.8 (71, 100) | 90.5 (71, 100) | 94.3 (88, 99) | 93.3 (87, 98) |
| Inf$_{sim}$ | 1 (0, 4) | 1 (0, 4) | 1 (0, 4) | 8 (2, 16) | 9 (3, 16) |
| P-value | $7.2 \times 10^{-1}$ | $8.1 \times 10^{-1}$ | $8.1 \times 10^{-1}$ | $7.1 \times 10^{-2}$ | $1.8 \times 10^{-1}$ |
| (traffic light: red, blue, green on) | (violin plot) | (violin plot) | (violin plot) | (violin plot) | (violin plot) |
| PrEP efficacy | 73.6 (21, 99) | 82.7 (53, 98) | 76.8 (44, 96) | 78.3 (66, 88) | 80.2 (69, 89) |
| Inf$_{sim}$ | 3 (0, 8) | 4 (0,9) | 5 (1, 10) | 33 (15, 58) | 28 (16, 43) |
| P-value | $3.8 \times 10^{-1}$ | $4.2 \times 10^{-1}$ | $2.9 \times 10^{-1}$ | $5. \times 10^{-5}$ | $1.6 \times 10^{-4}$ |

**Fig. 3 | Statistical assessment of pharmacological matrices to predict PrEP efficacy.** Each column corresponds to a clinical study. The top-most rows report the data-derived violin plot reporting uncertainty in the average efficacy when taken, its mean (95% confidence interval) and the mean number of infections simulated under trial conditions. The center and bottom rows report the corresponding results derived from mechanistic modeling (see Clinical Trial Simulations in Methods), testing the hypotheses indicated by the traffic light. The traffic light depicts the mechanistically modeled scenarios, from top-to-bottom: (i) If adherence was incomplete, drug concentrations in PBMC predicted efficacy and potency was estimated based on dNTP levels in CD4+ cells (red light on), vs. (ii) when efficacy was estimated based on drug concentrations in rectal tissue and potency was estimated based on dNTP levels in rectal tissue cells (red, green and blue lights on). For each hypothesis, a one-sided P-value was computed for differences in clinical trial outcomes (number of infected individuals) between the data-derived and mechanistically modeled hypotheses, under the null hypothesis ($\mathcal{H}_0$: distributions overlap) versus the alternative ($\mathcal{H}_1$: distributions do not overlap). P-values were estimated computed by running $10^5$ simulations per pair. Box plots show the median (white dot), interquartile range (box; 25th–75th percentiles), and whiskers extending to values within 1.5 x IQR, based on 1000 samples. Source data are provided as a Source Data file. PrEP: pre-exposure prophylaxis, PBMC: peripheral blood mononuclear cells, dNTP: deoxynucleoside triphosphate, IQR: interquartile range, Inf$_{sim}$: number of simulated infections.

## Adherence and PrEP efficacy

Based on the hypothesis testing (previous paragraph), we concluded that FTC-TP and TFV-DP pharmacokinetics in PBMC cells are predictive regarding the prophylactic efficacy of FTC/TDF-based oral PrEP in MSM. We therefore used our mechanistic model, with FTC-TP/TFV-DP pharmacokinetics in PBMC, to explore how rapid HIV protection was achieved after initiating PrEP and how long it persists after stopping PrEP both for daily PrEP, as well as PrEP on demand (Fig. 4). Note, that unlike graphics depicting instantaneous drug inhibition[37] our simulations predict PrEP efficacy, meaning that for each temporal gap $\Delta t$ = (time of virus exposure - time of first dose) our integrated pharmacology-viral dynamics model already considers how long drug concentrations have to be sustained to eventually clear all virus.

Based on our simulations, we observed that > 90% HIV risk reduction is achieved around the time of the first dose intake in a daily PrEP regimen (Fig. 4a), because the virus is cleared during subsequent dosing events. However, daily TDF/FTC-based oral prophylaxis is not efficient, if started after exposure, in line with previous work[38]. When daily PrEP is stopped, >90% HIV risk reduction persists for another 2 days. After that time, protection is variable, depending on inter-individual differences in the pharmacokinetic profiles as indicated by the shaded areas in Fig. 4a. For a 2-1-1 PrEP on demand regimen[15], we observe a similar onset of prophylactic efficacy, with -90% HIV risk reduction if virus exposure coincides with the first (double-)dose of PrEP, when followed by two more single dose events 24h apart. When PrEP on demand is stopped, >90% HIV risk reduction persists for another 24 hours (i.e. 72h after the first dose), after which protection is variable, depending on inter-individual differences in pharmacokinetics (Fig. 4b).

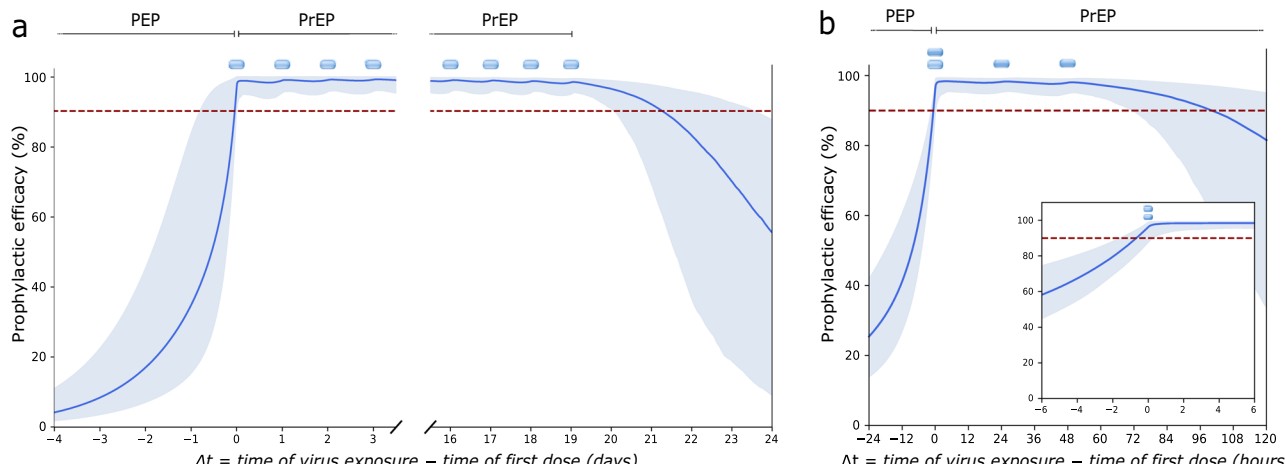

**Fig. 4 | Onset and persistence of PrEP efficacy. a** Onset of PrEP efficacy after initiation of daily oral TDF/FTC-based PrEP (dosing scheme indicated by blue 'pill' symbol) and persistence of HIV protection after stopping PrEP intake. **b** Onset of PrEP efficacy after 2-1-1 TDF/FTC-based PrEP on demand and duration of HIV protection after the last dose. Thick blue lines indicate the median PrEP efficacy and shaded areas correspond to 95% confidence intervals considering pharmacokinetic variability. Simulations were conducted assuming that FTC-TP and TFV-DP concentrations in PBMC predict prophylactic efficacy. Δ t = (time of virus exposure - time of first dose). Source data are provided as a Source Data file. TDF: tenofovir disoproxil fumarate, FTC: emtricitabine, PrEP: pre-exposure prophylaxis, FTC-TP: emtricitabine triphosphate, TFV-DP: tenofovir diphosphate, PEP: post-exposure prophylaxis, PBMC: peripheral blood mononuclear cells.

## Discussion

While a variety of drugs and drug formulations are becoming available for HIV prophylaxis, oral PrEP with TDF/FTC remains the most accessible and widely used option. However, it has been unclear to date, which pharmacokinetic matrix can be used to predict prophylactic efficacy for TDF/FTC-based oral PrEP. Knowledge of the relevant matrix may, on the other hand, enable predictions regarding the onset- and offset of HIV protection, and such knowledge can be highly valuable to AI-systems that support and improve self-management of HIV protection[39,40], with the overall prospect of increasing PrEP uptake and improving its use.[41]

In this work, we investigated whether drug concentrations in PBMC versus in local tissues predict prophylactic efficacy of oral TDF/FTC-based PrEP in MSM, based on a combination of clinical trial data analysis in MSM and mechanistic modeling. We found that two (HPTN 083 and DISCOVER) out of five clinical studies in MSM were statistically empowered to distinguish between these pharmacological matrices. Based on these studies we observed that concentrations in rectal tissues do not predict PrEP efficacy.

A major caveat of clinical trials with oral PrEP is the fact that drug intake is not supervised and consequently a proportion of study participants in the intervention arm may have interrupted-, stopped, or never taken the study drugs[42]. The proportion of non-takers may vary considerably between clinical trials (see Fig. 1a–e), complicating the comparison of PrEP efficacy across clinical trials. To allow estimating PrEP efficacy when used and enable comparison across trials, we corrected the observation time in the distinct studies for periods not covered by PrEP. Pharmacokinetic measurements of TFV in the blood plasma, as well as TFV-DP in DBS allow to detect individuals with no recent intake (plasma TFV) versus no long-term drug intake (TFV-DP in DBS).[43] The plasma TFV marker in isolation may be biased by white coat dosing (individuals taking the study drugs just before a clinical visit), which can lead to overestimation of individuals taking the drugs. Owing to the extraordinary long half-life, TFV-DP concentrations in DBS can be viewed as an account of past dosing, which may overestimate adherence for individuals who stopped taking PrEP. The combination of both markers can help to identify individuals who stopped taking PrEP (undetectable TFV and detectable TFV-DP in DBS), and may indicate white coat dosing effects (high plasma TFV and very low TFV-DP DBS concentrations), as outlined in Supplementary

Table 2. Notably, both markers are often only available for individuals who became infected in the trial, and single markers for randomly sampled individuals. Historically, plasma TFV was used as an adherence marker in early PrEP studies[15,21], while plasma TFV and TFV-DP in DBS was concurrently measured in later studies[27,28] and the most recent studies use DBS only[6] (an overview of available adherence markers is given in Supplementary Table 3). Since the assays used to assess adherence changed over time, adherence measurements from the distinct studies are not directly comparable. The two studies (HPTN 083 and DISCOVER) supporting our main findings allowed us to utilize the combination of both adherence markers in infected individuals. The combined use of both markers helped to identify 4/39 infected individuals with possible white coat dosing effects and 9/39 who likely stopped taking PrEP in HPTN 083 (Supplementary Table 3).

Based on the adherence markers, as well as their combination (Supplementary Table 2) we could identify that almost half of the individuals in the first PrEP study (iPrEX) did not take the drugs, whereas the fraction of non-drug taking individuals decreased to 4–21% in later studies (IPERGAY, HPTN 083, PURPOSE 2, DISCOVER) in MSM. Notably, PrEP efficacy in MSM was established in post-hoc analysis of the iPrEX study, which also (akin to our approach) dichotomized study participants based on whether FTC was detectable or not[21]. In stark contrast to MSM, in the recent PURPOSE 1 trial in women, only 7% and 16 % of individuals took oral TDF/FTC and TAF/FTC, respectively, at week 52 (Figure 3 in Bekker et al., 2024)[7], pointing at an urgent need to identify and overcome adherence barriers in this group.[44] On the other hand, the more recent clinical studies (IPERGAY, HPTN 083, PURPOSE2, DISCOVER) imply that MSM, while still facing obstacles, may not be affected by as many PrEP adherence barriers as cisgender women.

Based on the dichotomized clinical data, and considering study-specific adherence profiles (Supplementary Fig. 1), we quantified average PrEP-efficacy in MSM. Interestingly, while all studies pointed at high average PrEP efficacy in MSM who took any product, the HPTN 083[27] and DISCOVER[28] studies estimated PrEP efficacy with a narrow enough confidence interval that allowed discrimination between pharmacological matrices to predict PrEP efficacy. The statistical power of HPTN 083 is attributed to sufficient observation time (2685 person-years) in individuals taking the product, whereas in the DISCOVER trial, we have extraordinary observation time (4198 person-

years) in individuals taking the product. This explains why PrEP efficacy can be very confidently estimated from data of the DISCOVER trial, as well as HPTN 083. Our statistical analysis may also hint towards a mechanism by which counterfactual placebo controls[45], as used to evaluate the original PURPOSE 2 trial, may be verified using data from individuals not taking product in the oral-PrEP control arm of some studies. Moreover, once counterfactual placebo controls are established, individuals not taking oral PrEP could be excluded based on pharmacological markers and replaced by newly recruited study participants while a trial is running; which could optimize the tradeoff between the cost of a clinical trial and the deducible statistical information regarding PrEP efficacy.

In this work, using an entirely independent approach (mechanistic modeling of PrEP efficacy) that has been parameterized on disparate (pharmacological and viral dynamics) data, we could simulate the impact of pharmacological markers in different matrices on PrEP efficacy, and deduce PrEP adherence demands in MSM therefrom. These computational experiments assess whether drug concentrations in tissue-resident CD4+ cells correlate (i) with the tissue they are embedded in (= concentrations in tissue are relevant), or (ii) the cell type they belong to (= concentrations in PBMC are relevant). When considering colorectal tissue pharmacokinetics (local drug concentrations) even fully adherent individuals would not surpass 90% HIV risk reduction (Fig. 2). Mechanistically, this low efficacy is due to the fact that FTC-TP concentrations in colorectal tissue are substantially reduced compared to PBMC (see Methods section). Overall, when using colorectal tissue drug pharmacokinetics in MSM, predicted PrEP efficacy was higher than corresponding efficacy in cisgender women than when utilizing vaginal tissue pharmacokinetics (Fig. 2)[24]. Notably, utilization of colorectal tissue pharmacokinetics in the mechanistic model would not support PrEP on demand for MSM, as efficacy starts to drop-off for 4 or less pills per week (Fig. 2c). In contrast, when we used drug pharmacokinetics in PBMC to predict PrEP efficacy in the mechanistic model, > 95% protection was achieved in fully adherent individuals, and 3–4 pills per week would still reduce HIV infection risk by >90% in over 75% of all individuals (Fig. 2d). Thus, drug pharmacokinetics in PBMC, but not colorectal tissues, support the PrEP on demand regimen in MSM.

Colorectal tissue homogenates include many cell types not susceptible to HIV infection[46] with vastly different FTC-TP and TFV-DP uptake and phosphorylation kinetics, in comparison to CD4+ cells.[47,48]. Thus, TFV-DP and FTC-TP concentrations in colorectal tissue biopsies may not accurately reflect drug concentrations in CD4+ T-cells relevant to the infection. Concentration measurements in CD4+ T-cells at the colorectal exposure site are unfortunately difficult to obtain and largely missing, except for some animal studies measuring TFV-DP at these sites after topical dosing[49]. On the other hand, PBMC consist primarily of CD4+ T cells, and it has indeed been shown that FTC-TP and TFV-DP concentrations in CD4+ T cells in PBMC correlate well with concentrations in CD4+ T cells in general.[50–52]. Notably, drug concentrations in PBMC have been well-established as markers for systemic response and more directly translating to efficacy.[27,42,53,54]. Therefore, the cell type, rather than the physical locations, may be a determinant of TDF/FTC-based oral PrEP efficacy. Lastly, during oral PrEP, FTC and TFV may enter local exposure sites (and resident CD4+ cells) via the systemic circulation, where they get taken up and phosphorylated into their active form in a cell-specific manner. It may therefore be plausible that drug concentrations in systemic PBMC correlate with drug concentrations in exposure-site resident CD4+ cells. Ultimately, in order to probe the conflicting hypotheses regarding suitable pharmacological matrices, we simulated the distinct PrEP trials, utilizing trial-specific adherence behavior to compute adherence-averaged PrEP efficacy in individuals with detectable drug, akin to a previous work focusing on cisgender women.[24]. We then statistically compared derived efficacy distributions. Notably, for the

distinct clinical trials, utilizing pharmacokinetics in colorectal tissue and trial-specific adherence profiles resulted in central estimates of average HIV risk reduction between 70- and 80%, whereas simulations considering pharmacokinetics in PBMC yielded estimates >90%. Interestingly, only data from the HPTN 083 and DISCOVER trial would allow to statistically distinguish between these mechanistically-derived estimates, because of the tight confidence interval around the average PrEP efficacy estimate of 96.5% (CI: 93-99%) and 98.7% (CI: 97-99%) obtained from clinical data. In fact, HPTN 083 and DISCOVER were comparing different PrEP agents (TDF/FTC vs. injectible CAB and TDF/FTC vs. TAF/FTC). Since both arms of the original studies included a form of PrEP, HIV incidences were expected to be low and required substantial observation time to ascertain reduction in HIV incidence between the two investigated PrEP interventions.[55] This statistical empowerment of DISCOVER and HPTN 083 allowed our re-analysis to deduce statistically strong PrEP efficacy estimates.

Taken together, our combined approach demonstrates that TFV-DP and FTC-TP concentrations in colorectal tissue homogenates do not predict PrEP efficacy in MSM. Notably, our simulations assumed that drug concentrations at the site of effect (in tissue-resident CD4+ cells) are either predicted by concentrations in rectal tissue, or that they can be predicted by drug concentrations in PBMC. In the Supplementary Text and Supplementary Table 4 we also computed effect-site concentrations from a combinations of both matrices. These simulations indicated that if effect site concentrations are predicted from 10% rectal tissue (the remaining proportion being PBMC), the resultant clinical trial simulations are incongruent with the DISCOVER trial. If the tissue contribution is further increased to ≥50%, predictions become incompatible with both informative trials (HPTN 083 and DISCOVER).

Simulations based on drug levels in PBMC, suggested that > 2 doses per week may achieve > 90% protection in the majority ( ≈ 75%) of MSM. Our efficacy estimates using PBMC support on-demand PrEP in MSM and are broadly in agreement with recent work, where HPTN 083 trial participants were further stratified into adherence categories,[56] although the stratification in ref. [56] may be statistically underpowered in the lower adherence strata (<2 and 2–3 doses per week; see Supplementary Table 2 for details on pharmacokinetic markers for infected individuals in HPTN 083). Moreover, our results strongly support earlier works that established adherence-efficacy relationships from PBMC levels measured in the iPrEX study.[50] Because we had previously demonstrated that concentrations in vaginal tissue homogenates do not predict PrEP efficacy in cisgender women[24], taken together, our two studies question the mechanistic basis for distinguishing PrEP adherence requirements in MSM and cisgender women, as currently specified in the WHO guidelines.

While our work indicated that there may not be differences between MSM and cisgender women with regards to adherence requirements (Fig. 2d), there are undeniable differences in PrEP adherence behavior between cisgender women and MSM: I.e., in the recent PURPOSE 1[7] (cisgender women) vs. PURPOSE 2 trial (MSM), only 7% vs. 73% participants had drug levels consistent with ≥2 doses a week at trial-week 52. Comparative studies among trials like iPrEx (MSM), Partners PrEP (heterosexual couples), and VOICE (cisgender women) suggest that adherence varies by gender and risk level. For example, while the Partners PrEP study showed low incidence and higher adherence among women (median TFV concentration 80 ng/mL)[22], VOICE had higher incidence rates (6.9 infections per 1000 person-years) with poor adherence in women at higher risk.[18] This difference underscores how risk context affects adherence.[57] Together, these findings highlight how contextual and behavioral factors impact PrEP acceptability and willingness to take PrEP in women, indicating the need for refined approaches in risk assessment, adherence measurement in PrEP studies[57,58], as well as the importance of measures to support PrEP uptake, particularly in cisgender women.[59–61]

Lastly, based on the PBMC efficacy marker, we evaluated the onset of PrEP protection and its waning after stopping PrEP. For both daily PrEP and PrEP-on demand, we observed a rapid onset of >90% protection, within the first hour after the first dose and lasting 53 (24, 104) hours after the last dose (median and 95% confidence interval based on simulated population, see Fig 4). Our simulations therefore support current guidelines on the use of on-demand PrEP[12], which should be initiated within 2-24h before virus exposure. Moreover, our simulations indicate that >90% protection is even achieved if the 2-1-1 cycle was completed 24h before virus exposure (Fig. 4).

## Limitations and Further Research

A potential limitation of our work is the fact that we used previously developed pharmacokinetic models that were derived from studies in women.[33,34] Previous studies[62–64] indicated that there are no distinct gender differences between PK parameters for TFV in blood plasma and PBMC. To verify this assumption, we re-analyzed data from the DOT-DBS study[43] stratified by gender finding no significant differences (Supplementary Table 1).

When dichotomizing the clinical trial intervention arms, we generalize adherence measurements from a subset of individuals to the whole intervention arm. While this is due to the limited testing capacity in clinical trials, it may lead to potential over- or underestimation of observation time (person-years) in each group which in turn affects estimates of infection incidence. The dichotomization of observation time (person-years) into drug-taking vs. not-taking represents a coarse method of data separation. However, introducing more adherence categories reduces observation time and the number of infections per category, which negatively impacts on statistical power. In addition, we could not rigidly use the same criteria across all 5 studies due to differences in availability of participant level drug concentrations (versus binned adherence data) plus a variety of analytes, biological matrices, and assay sensitivity. In contrast to other works[56], we aim to maximize statistical power by estimating average PrEP efficacy across different adherence strata in the respective clinical trials. Some analyzed trials (HPTN 083, PURPOSE 2, DISCOVER) did not contain placebo arms, which is necessary to deduce a baseline incidence, whose reduction relates to PrEP efficacy. In this work, we instead used the subgroup of individuals with undetectable drugs as a placebo control within the intervention arm (Fig. 1e). To circumvent over-confidence in the deduced incidence estimates, we inversely sampled incidence rates derived through this method (see Methods), which enables to represent statistical uncertainty in placebo-like incidence rates throughout the modeling workflow. Other groups address the problem of missing placebo arms by developing statistical approaches focusing on averted HIV infections.[55,65] In those approaches, infection incidence is estimated from previous clinical trials. Our analysis (Fig. 1e) highlights that background incidence may however vary between clinical trials and therefore points at a potential difficulty in transferring incidence rates from one study population to another. Recent developments in estimating incidence in counterfactual placebo controls may however provide a means to enable combining different study populations for concomitant analysis.[45]

Lastly, while our mechanistic modeling with drug pharmacokinetics in PBMC highlights that at least 3 doses per week may provide > 90% HIV risk reduction (Fig. 2d), our modeling also supports current WHO recommendations for on demand PrEP in MSM[12] (Fig. 4b).

## Methods

### PrEP trials for TDF/FTC-based oral PrEP in MSM

We analyzed publicly available data from iPrEX, HPTN 083, IPERGAY, DISCOVER and PURPOSE 2[6,15,21,27,28]. All primary endpoints for these trials have been published and ethical approvals and written informed consent were obtained. In brief, iPrEX tested the efficacy of daily oral TDF/FTC-based PrEP vs. a placebo of which 1217 MSM were in the TDF/

FTC PrEP intervention arm (1628 person-years of observation time). A total of 64 vs. 34 new infections were recorded in the placebo and intervention arm. TFV plasma concentrations was measured in 43 random samples to assess overall adherence in the intervention arm, and was measured in 34 individuals who became infected in the intervention arm. TFV was detectable in 51% of the the random samples, but only in 9% of individuals who became infected.

HPNT 083 assessed long-acting PrEP with injectable carbotegravir vs. oral TDF/FTC-based daily PrEP (control arm) in 2247 MSM and transgender women (TGW)[27] amounting to 3123 person-years of observation time. Adherence in the TDF/FTC control group was assessed both in terms of plasma TFV (acute adherence) and TFV-DP in dried blood spots (long-term adherence marker) based on a total of 390 samples. Plasma TFV was detectable in 86% of the random samples and 6/39 (15%) infected individuals had evidence of recent drug intake in the TDF/FTC arm of the study (see also Supplementary Table 2.).

IPERGAY tested on-demand PrEP in 199 MSM vs. a placebo in 201 MSM, contributing to 221 vs. 223 person-years of observation time. 14 vs. 2 infections were observed in the placebo vs. intervention arm. Adherence was measured using plasma TFV and participants reported using a median of 15 tablets per month[15]. A total of 86% of individuals had detectable TFV in the intervention arm and 0% of those infected in the intervention arm had detectable plasma TFV.

DISCOVER assessed TAF/FTC-based daily oral PrEP with vs. oral TDF/FTC-based daily oral PrEP (control arm) in 5387 MSM amounting to 8756 person years observation time[28]. Adherence in the TDF/FTC control group was assessed both in terms of plasma TFV (acute adherence) and TFV-DP in dried blood spots (long-term adherence marker) in a randomly pre-selected subset of 536 participants. In the TDF/FTC arm of the study, TFV-DP was detectable in all sampled individuals, including those who became infected. Notably, 96% of the participants had TFV-DP levels consistent with ≥2 doses per week. Among the infected individuals, only one (9%) showed levels consistent with recent drug intake (≥4 doses per week), while the remaining had levels indicating <2 doses per week (91%). PURPOSE 2 assessed long-acting PrEP with injectable lenacapavir vs. oral TDF/FTC-based daily PrEP (control arm) in MSM and transgender women (TGW)[6], of which 920 individuals were included in the TDF/FTC-based arm, amounting to 828 person-years of observation time in the FTC/TDF control arm. A total of 8 new infections were observed in the FTC/TDF control arm. Adherence was assessed through TFV-DP concentrations in red blood cells (RBC). According to the TFV-DP levels detected in RBCs, a total of 20% individuals took one dose per week or less, which corresponds to the adherence strata used for plasma TFV[24] (Supplementary Fig. S1 therein). Of the 8 infected individuals, 2 individuals had quantifiable TFV-DP levels.

**Incidence rate estimation.** HIV incidence is typically calculated as $r_{Inf} = \frac{\text{number of infected individuals}}{\text{observation time}}$. However, because the observed number of infections tends to be low in a given study, incidence estimates calculated directly from clinical data are statistically uncertain. We utilized a previously developed method[24] to capture this uncertainty. In brief, we assumed the number of observed infections ($N_{Inf}$) during a clinical trial to be binomially distributed and then inverse sampled incidence parameters using the simulation_utils.py module from https://github.com/KleistLab/PrEP_Truvada/. To sample incidence rates from the respective clinical trials, we utilized the number of individuals in the sub-group of individuals with no plasma TFV detectable, the observation time and the number of observed infections from each trial, compare Fig. 1e.

### Estimation of prophylactic efficacy from clinical data

We adopted a Bayesian approach to estimate prophylactic efficacy $\varphi \in [0, 1]$ (HIV risk reduction) in the subgroup of individuals who took (some) drug. The posterior probability of PrEP efficacy $\varphi$ after

observing $\text{Inf}_{\text{obs}}$ infections is defined as:

$$P(\varphi|\text{Inf}_{\text{obs}}) = \frac{P(\text{Inf}_{\text{obs}}|\varphi) \cdot P(\varphi)}{P(\text{Inf}_{\text{obs}})}, \quad (1)$$

In the following, we assume a uniform prior regarding PrEP efficacy, $P(\varphi) = \mathcal{U}(0,1)$. We model the likelihood of observing a particular number of infections assuming a binomial distribution:

$$P(\text{Inf}_{\text{obs}}|\varphi) = \mathcal{B}(\text{Inf}_{\text{obs}}; N_{\text{tot}}, r_{\text{inf}}(\varphi)), \quad (2)$$

where $N_{\text{tot}}$ denotes the number of individuals in the drug detected subgroup, $r_{\text{inf}}(\varphi) = r_{\text{inf}}(\varnothing) \cdot (1 - \varphi)$ is the infection rate for PrEP efficacy $\varphi$ and $r_{\text{inf}}(\varnothing)$ is the baseline infection rate in the placebo group (or the subgroup of the intervention arm where the drug was undetectable). The marginal likelihood in the denominator ensures proper normalization:

$$P(\text{Inf}_{\text{obs}}) = \int_0^1 P(\text{Inf}_{\text{obs}}|\varphi) \cdot P(\varphi) \, d\varphi. \quad (3)$$

We compared this approach to a previously developed approach[24] (online Methods section therein) that utilizes the number of observed infections in the entire intervention arm (with- and without detectable drug). This method confirmed all findings presented in this manuscript.

**Information gain**. To quantify, whether a clinical trial was informative for quantifying average PrEP efficacy, we computed information gain as

$$I(\varphi) = H_{\text{max}} - H(\varphi) \quad (4)$$

where $H_{\text{max}} = \log_2(100)$ denotes the Shannon Entropy of a discrete uniform distribution on the interval $[0, 100]$ (=our prior regarding the prophylactic efficacy $\varphi$ in %) and $H(\varphi) = \int_0^{100} P(\varphi) \log_2(P(\varphi)) \, d\varphi$ denotes the Shannon Entropy of $P(\varphi)$ derived from the clinical data. I.e., in case of $\varphi_i \sim \mathcal{U}(0, 100)$, $H(\varphi) = H_{\text{max}}$ and there was no information gain. In contrast, if $P(\varphi_i)$ was a point distribution, $H(\varphi) = 0$ and therefore $I(\varphi) = H_{\text{max}}$.

## Pharmacokinetics of Oral TDF/FTC

The pharmacokinetic (PK) model was adopted from the models by Burns et al. (2015)[33] and Garrett et al. (2018)[34]. In brief, FTC and TDF are dosed into respective dosing compartments $D$ and absorbed into the blood plasma with rate parameter $k_a$. During this process TDF is converted into its circulating form tenofovir (TFV) by first-pass metabolism. The circulation of TFV and FTC in blood plasma- ($A_1$) and peripheral ($A_2$) compartments are described by first-order reactions with rate parameters $k_{12}$ and $k_{21}$ depending on the direction of flux. TFV and FTC can be taken up from the plasma, and be intracellularly phosphorylated into their active moieties in PBMC (compartment $A_3$) with rate functions $f_{13}$ and $f_{31}$ depending on the direction of flux. For FTC, nonlinear uptake and conversion kinetics with $f_{13} = \frac{V_{max} \cdot A_1}{K_m + A1}$ and linear efflux kinetics were considered with $f_{31} = k_{31} \cdot A_3$. For TFV, linear uptake kinetics were considered $f_{13} = k_{13} \cdot A_1$, as well as intracellular elimination $f_{30} = k_{31} \cdot A_3$.

$$\frac{d}{dt}D = -k_a \cdot D \quad (5)$$

$$\frac{d}{dt}A_1 = k_a \cdot D - k_{12} \cdot A_1 + k_{21} \cdot A_2 - k_e \cdot A1 - f_{13}(A_1) + f_{31}(A_3) \quad (6)$$

$$\frac{d}{dt}A_2 = k_{12} \cdot A_1 - k_{21} \cdot A_2 \quad (7)$$

$$\frac{d}{dt}A_3 = f_{13}(A_1) - f_{31}(A_3) - f_{30}(A_3) \quad (8)$$

Calculations were performed with molecular units to avoid conversion factors that account for the distinct masses of the drug metabolites.

To reflect pharmacokinetic variability, we utilized parameters from 1000 virtual individuals as previously described[24] (Supplementary Data sets therein) and simulated pharmacokinetic trajectories for the four compartments and any given adherence pattern over a time span of nine month.

**Gender differences with regards to PK**. Previous studies indicated that no gender differences existed with regards to PK parameters[62–64]. For verification, we developed two population pharmacokinetic (Pop-PK) models to evaluate sex effects on TDF and FTC PK using data from the DOT-DBS study[43] and the HPTN 066 study[63]. The Pop-PK analysis was performed using NONMEM version 7.5 (ICON Development Solutions, MD, USA) with Perl speaks NONMEM (PsN version 4.9.0) and Pirana as the interfaces. Data visualization, NONMEM dataset preparation, and NONMEM post-run diagnostics, and figure preparation were conducted using R version 4.3.2 (R Development Core Team; http://www.r-project.org/) and R studio (2024.09.1+394). All TFV and FTC plasma concentrations were converted to molar concentrations with its molecular weight (TFV: 287.2 g/mol, FTC: 247.25 g/mol). TFV-dp and FTC-tp concentrations in PBMC were converted from fmol/million cells to umol/L based on the volume of a single PBMC (282 fL/cell). Handling of below-the-limit-of-quantification (BLQ) data varied by analyte. For plasma TFV and FTC concentration, BLQ data were handled using the Beal M3 method. For TFV-dp and FTC in PBMC samples, where each sample had a unique lower limit of quantification (LLOQ), the Beal M1 method was applied to handle BLQ data. We first modeled TFV and FTC in plasma, followed by incorporation of a PBMC compartment into the base model. Model performance was assessed using diagnostic plots, including observed versus population and individual predicted concentrations and conditional weighted residuals versus predicted concentrations or time. The objective function value (OFV) was used for model comparison. Internal validation was performed using visual predictive checks (VPCs). Overall, this analysis concluded with no sex differences detectable as shown in Supplementary Table 1.

**Pharmacokinetics in exposed tissue**. In a previous study[24], we used all available data (16 individual studies) that measured TFV-DP and FTC-TP concentrations in local tissue, as well as in PBMC. By modeling the dosing schedules from each of these studies, we could deduce a ratio of drug levels between local tissue and PBMC that appropriately captured pharmacokinetics in local tissue. For TFV-DP we deduced that concentrations in colorectal tissue homogenates were 2.92 times greater than the concentration in PBMC, whereas FTC-TP concentrations in colorectal tissue were 0.04 times as high as the corresponding concentrations in PBMC. For comparisons between colorectal and vaginal tissues in Fig. 2c we used previously derived vaginal tissue-to PBMC ration of 0.07 and 0.06 for TFV-DP and FTC-TP respectively.

## Drug potency & pharmacodynamics

We utilized a previously developed[36] and validated[66] molecular mechanisms of action (MMOA) model to estimate drug potency ($IC_{50}$), as well as to model the synergistic inhibition by TFV-DP and FTC-TP[32], expressed in terms of their concomitant direct drug effect $\eta(I_1, I_2)$. Notably, this model takes deoxynucleoside triphosphate (dNTP) levels at the site of action into account. Hence we could use the model with dNTP levels measured in CD4+ cells vs. in rectal tissue cells[23] to predict

drug potency if either PBMC were a marker of efficacy (consisting mainly of CD4+ cells), or if rectal tissue was the effect site, as previously described.[24]

## Virus dynamics model and prophylactic efficacy

We utilized the HIV dynamics model from[29], which takes into account free viruses ($V$), early infected T cells ($T_1$) and late infected T cells ($T_2$). In brief, the HIV replication cycle is modeled by 6 reactions $R_1 - R_6$ with propensities $a_1-a_6$ and stoichiometry as outlined below.

$$R_1 : V \rightarrow \varnothing ; \quad a_1(I_1, I_2) = CL + \left( \frac{1}{\rho_{rev, \varnothing}} - (1 - \eta(I_1, I_2)) \beta \cdot T_u \right) V \quad (9)$$

$$R_2 : T_1 \rightarrow \varnothing ; \quad a_2 = (\delta_{PIC} + \delta_{T_1}) T_1 \quad (10)$$

$$R_3 : T_2 \rightarrow \varnothing ; \quad a_3 = \delta_2 T_2 \quad (11)$$

$$R_4 : V \rightarrow T_1; \quad a_4(I_1, I_2) = (1 - \eta(I_1, I_2)) \beta \cdot T_u \cdot V \quad (12)$$

$$R_5 : T_1 \rightarrow T_2; \quad a_5 = k \cdot T_1 \quad (13)$$

$$R_6 : T_2 \rightarrow V + T_2; \quad a_6 = N_T \cdot T_2 \quad (14)$$

where the first three reactions describe the clearance of free virus by the immune system with rate parameter CL or by unsuccessful infection with parameter $\left( \frac{1}{\rho_{rev, \varnothing}} - (1 - \eta(I_1, I_2)) \beta \cdot T_u \right)$, the clearance of the pre-integration complex $\delta_{PIC}$ or early infected cells with parameter $\delta_1$, and the clearance of late infected cells ($T_2$) with parameter $\delta_2$. The last three reactions describe the infection of a previously uninfected T-cell $T_u$ by free virus with rate parameter $(1-\eta(I_1, I_2)) \beta \cdot T_u \cdot V$, the propagation of an early-infected cell to a late infected cell by proviral integration and cell reprogramming with rate parameter $k$, and the generation of novel viruses from late infected cells with rate parameter $N_T$. In the equations above, $\eta(I_1, I_2)$ denotes the direct effect of TFV-DP ($I_1$) and FTC-TP ($I_2$), depending on their target site pharmacokinetics (either local tissue or PBMC). All other parameters were taken from Zhang et al. (2021) (Table 1 therein).[35]

We calculated the prophylactic efficacy per exposure from the viral dynamics model in terms of the relative reduction in infection probability per exposure for a prophylactic regimen $S$ compared to the absence of drugs $\varnothing$ after virus challenge $Y$ at time $t$ ($Y_t$).[35]

$$\varphi(Y_t, S) = 1 - \frac{P_I(Y_t, S)}{P_I(Y_t, \varnothing)}, \quad (15)$$

where $P(Y_t, \varnothing)$ was calculated analytically.[31] The variable $P_I(Y_t, S)$ was calculated from its complement extinction probability $P_I(Y_t, S) = 1 - P_E(Y_t, S)$. The latter was computed using the method developed in ref. 35 (Probability Generating System) which, for the considered virus dynamics model, results in the following system of coupled ODEs, which is solved backwards in time by considering how an individual pharmacokinetic trajectory affects reactions $a_1$ and $a_4$ through time. The backwards integration is performed in matrix form, such that solutions for $N$ (here: $N = 1000$) virtual patients for a given adherence profile are computed simultaneously using standard ODE solvers.

$$\frac{dP_E(Y_t = \widehat{V})}{dt} = a_1(t) \cdot (P_E(Y_t = \widehat{V}) - 1) + a_4(t) \cdot (P_E(Y_t = \widehat{V}) - P_E(Y_t = \widehat{T_1})) \quad (16)$$

$$\frac{dP_E(Y_t = \widehat{T_1})}{dt} = a_2 \cdot (P_E(Y_t = \widehat{T_1}) - 1) + a_5 \cdot (P_E(Y_t = \widehat{T_1}) - P_E(Y_t = \widehat{T_2})) \quad (17)$$

$$\frac{dP_E(Y_t = \widehat{T_2})}{dt} = a_3 \cdot (P_E(Y_t = \widehat{T_2}) - 1) + a_6 \cdot (P_E(Y_t = \widehat{T_2}) - P_E(Y_t = \widehat{T_2} \cdot P_E(Y_t = \widehat{V}))) \quad (18)$$

In this system of ODEs, $P_E(Y_t = \widehat{V})$, $P_E(Y_t = \widehat{T_1})$ and $P_E(Y_t = \widehat{T_2})$ denote the probability of viral extinction if only one virus, one early infected cells, or one late infected cells were present. The probability of extinction for any possible state of the viral dynamics model is calculated by assuming statistical independence, i.e.

$$P_E(Y_t) = P_E(Y_t = \widehat{V})^{\#V} \cdot P_E(Y_t = \widehat{T_1})^{\#T_1} \cdot P_E(Y_t = \widehat{T_2})^{\#T_2} \quad (19)$$

where $\#V$, $\#T_1$ and $\#T_2$ denotes the initial number of viruses, early- and late infected T-cells.

**Receptive Anal Virus Exposure.** Modeling of receptive anal intercourse (RAI) was implemented in this system by considering the exposure-specific inoculum size[67], see also ref. 24 (Supplementary Fig. S10 therein) and by either utilizing FTC-TP & TFV-DP pharmacokinetics and pharmacodynamics in rectal tissue or PBMC, according to the employed hypothesis testing scheme.

**PrEP adherence.** In the HPTN 083, PURPOSE 2, and DISCOVER trial, adherence was directly quantified using TFV-DP concentrations in red blood cells (RBCs). For these trials, we assumed a uniform distribution within the individual adherence categories stated in the original studies (e.g., 3-5 doses/week), as shown in Supplementary Fig. 1. We then computed adherence-averaged PrEP efficacy directly utilizing the adherence-efficacy relationships depicted in Fig. 2c, d. For IPER-GAY and iPrEx, only plasma TFV levels were available and we utilized our pharmacokinetic models to compute the probability of TFV plasma levels above the lower limit of quantification (LLOQ; compare Supplementary Fig. S1, 2 in ref. 24) for different adherence strata (number of pills per week). This allowed us to estimate adherence frequencies for these studies as well, as shown in Supplementary Fig. 1.

## Clinical Trial Simulation

Outcomes from actual clinical trials (i.e., number of infected individuals) may be subject to intrinsic stochasticity. I.e., hypothetically repeating the same trial may yield distinct outcomes since there is randomness in whether an infection may occur before vs. after the end of the trial's observation period or patient drop-out. To account for this intrinsic randomness, we simulated clinical trials using Gillespie's algorithm.[68] The trial is modeled by two possible events: either an individual is infected during the observation period or drops out (= observation period ended). These events occur with reaction rates; $r_{Inf}$ and $r_{dr-out}$ and stoichiometries:

$$R_1 : S \xrightarrow{r_{Inf}} I \quad \text{(infection event)} \quad (20)$$

$$R_2 : S \xrightarrow{r_{dr-out}} \varnothing \quad \text{(drop-out or end of observation)} \quad (21)$$

where $I$ and $S$ denote the infected individuals and the remaining number of susceptible individuals and respectively. To simulate the drug-detected group in the data-driven approach, we assigned an individual-level efficacy value $\varphi$ to each simulated individual from the PrEP efficacy distributions derived from Bayesian inference as discussed above. Accordingly, we utilized an adapted infection incidence $r_{Inf}(\varphi) = r_{Inf} \cdot (1 - \varphi)$, where $r_{Inf}$ is the infection incidence in the absence of drug. We simulated each trial 100,000 times by inverse

sampling the incidence rate and computing $r_{dr-out} = 1/\widehat{T} - r_{Inf}(\varphi)$, where $\widehat{T}$ denotes the average follow-up time per individual.

When testing the mechanistic hypotheses, we sampled the simulated time-averaged individual-level efficacy $\varphi$ from our virtual population (compare Fig. 2). This efficacy was sampled according to the probability of detecting TFV or, when available, reported adherence data in the trial (see Supplementary Fig. 1).

**Hypothesis testing.** Lastly, the PrEP efficacy estimates from clinical data were used to evaluate the mechanistic hypotheses, as previously described in Zhang et al., 2023.[24] In brief, we computed a one-sided *P*-value as the frequency by which the empirically computed distributions of the number of infected individuals at the end of the respective trials overlapped (100,000 simulations respectively). Note that this denotes a conservative test that compared distributions instead of central values (means, medians).

### Reporting summary
Further information on research design is available in the Nature Portfolio Reporting Summary linked to this article.

## Data availability
The datasets used in the analysis, along with processed simulation data are available in the published code at https://github.com/KleistLab/PrEP_Truvada with https://zenodo.org/records/19147627[69]. Source data are provided with this paper.

## Code availability
All programs have been implemented in Python 3 using the SciPy and the matplotlib package and are freely available at https://github.com/KleistLab/PrEP_Truvada with https://zenodo.org/records/19147627[69].

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

## Acknowledgements

L.Z., S.I. and M.v.K. thank the German Ministry for Research, Technology, and Space (grant no. 01KI2016), as well as the German Ministry for Health (grant no. 2524FSB62A) for funding. M.v.K. acknowledges funding provided through the German Research Foundation excellence center Math+. The funders had no role in the design of the study or the decision to publish.

## Author contributions

S.I., M.M., C.W.H. and M.v.K. conceived the project. Y.Y., L.Z., and R.R.B. provided methodology. S.I., M.M. and Y.Y. performed investigations. S.I., M.M. and M.v.K. wrote the original draft of the manuscript. R.R.B. and M.v.K. acquired the funding. M.v.K. supervised the project.

## Funding

## Competing interests

The authors declare no competing interests.
