## [Peer Review file · Nature Communications]

Pharmacological markers of HIV prevention for oral pre-exposure prophylaxis in Men who have Sex with Men

Corresponding Author: Professor Max von Kleist

Version 0:

Reviewer comments:

Reviewer #1

(Remarks to the Author)

With their response to the reviewer comments and additional work, the authors have adequately responded to the concerns and substantially strengthened and clarified the paper. It is clearer what was done why, particularly where it deviates from less mathematically intense terminology. I especially appreciate the clarification on the figures and reworking of the axes/adding explanation there. I have no further comments to address.

(Remarks on code availability)

Reviewer #2

(Remarks to the Author)

I think the authors addressed appropriately reviewer's comments in this revision. I have one suggestion. The authors may wish to consider macaque data from the CDC group that detailed PrEP efficacy and associated drug levels in rectal and vaginal tissues from topically applied gels containing tenofovir (Dobard et al J Virol 2012, Dobard et al JID 2015). These data helped define the contributions of drugs in tissues to protection. The data showed that topical tenofovir dosing is highly protective but requires substantially higher (>100-fold) TFV-DP drug concentrations in tissues than what is afforded by oral TDF. The findings in this model suggest a possibly minor role of tissue drugs in protection by oral PrEP and point to the systemic drug levels playing a bigger role in PrEP efficacy, and possibly more relevant for predicting efficacy. The authors may wish to discuss these data if they find them helpful.

(Remarks on code availability)

We made final changes to the discussion corresponding to the remaining suggestion by reviewer #2 and made the appropriate changes corresponding to the provided checklist.

Remaining comment:

Reviewer #2 (Remarks to the Author):

I think the authors addressed appropriately reviewer's comments in this revision. I have one suggestion. The authors may wish to consider macaque data from the CDC group that detailed PrEP efficacy and associated drug levels in rectal and vaginal tissues from topically applied gels containing tenofovir (Dobard et al J Virol 2012, Dobard et al JID 2015). These data helped define the contributions of drugs in tissues to protection. The data showed that topical tenofovir dosing is highly protective but requires substantially higher (>100-fold) TFV-DP drug concentrations in tissues than what is afforded by oral TDF. The findings in this model suggest a possibly minor role of tissue drugs in protection by oral PrEP and point to the systemic drug levels playing a bigger role in PrEP efficacy, and possibly more relevant for predicting efficacy. The authors may wish to discuss these data if they find them helpful.

=> Of the two suggested works on macaque data, we included the one TFV-DP concentrations in colorectal tissues and local lymphocytes (Dobard et al JID 2015) in the discussion.